# Factorial Design Statistical Analysis and Optimization of the Adsorptive Removal of COD from Olive Mill Wastewater Using Sugarcane Bagasse as a Low-Cost Adsorbent

Fatima Elayadi [1,2], Mounia Achak [2,3,*], Wafaa Boumya [4], Sabah Elamraoui [2], Noureddine Barka [4], Edvina Lamy [5], Nadia Beniich [6] and Chakib El Adlouni [1]

1 Laboratory of Marine Biotechnologies and Environment, Sciences Faculty, Chouaïb Doukkali University, El Jadida 24000, Morocco
2 Science Engineer Laboratory for Energy, National School of Applied Sciences, Chouaïb Doukkali University, El Jadida 24000, Morocco; sabahamraoui0@gmail.com
3 Chemical & Biochemical Sciences, Green Process Engineering, CBS, Mohammed VI Polytechnic University, Ben Guerir 43150, Morocco
4 Multidisciplinary Research and 12 Innovation Laboratory, FP Khouribga, Sultan Moulay Slimane University of Beni Mellal, Beni-Mellal 23000, Morocco
5 EA 4297 TIMR Laboratory, Department of Chemical Engineering, Centre de Recherche de Royallieu, University of Technology of Compiegne—Sorbonne University, Rue du Dr Schweitzer, CS 60319, 60200 Compiegne, France
6 Dynamical Systems Laboratory, Sciences Faculty, Chouaïb Doukkali University, El Jadida 24000, Morocco
* Correspondence: achak_mounia@yahoo.fr; Tel.: +212-6614-74231

**Abstract:** This work highlights the elimination of chemical oxygen demand (COD) from olive mill wastewater using sugarcane bagasse. A $2^{5-1}$ fractional factorial design of experiments was used to obtain the optimum conditions for each parameter that influence the adsorption process. The influence of the concentration of sugarcane bagasse, solution pH, reaction time, temperature, and agitation speed on the percent of COD removal were considered. The design experiment describes a highly significant second-order quadratic model that provided a high removal rate of 55.07% by employing optimized factors, i.e., a temperature of 60 °C, an adsorbent dose of 10 g/L, a pH of 12, a contact time of 1 h, and a stirring speed of 80 rpm. The experimental data acquired at optimal conditions were confirmed using several isotherms and kinetic models to assess the solute interaction behavior and kind of adsorption. The results indicated that the experimental data were properly fitted with the pseudo-first-order kinetic model, whereas the Langmuir model was the best model for explaining the adsorption equilibrium.

**Keywords:** adsorption; sugarcane bagasse; olive mills wastewater; factorial design

## 1. Introduction

Olive mill wastewater (OMW) is an important environmental problem due to the strong color, low pH, and high concentrations of chemical oxygen demand (COD), biochemical oxygen demand (BOD), and phenolic compounds [1,2]. The discharge of this effluent into soil or rivers without treatment leads to damages to the environment that are attributed to their potential risk to flora or depletion of clean water reservoirs [3]. Hence, the treatment of OMW is a great challenge in a problematic environment. There are numerous approaches that can be used to remove organic pollutants and reduce COD levels from OMW such as coagulation–flocculation, aerobic and anaerobic biodegradation, solar distillation, adsorption, and infiltration percolation [4–10]. Among these techniques, adsorption is a well-established technique that benefits from advantages such as a simple design, low-cost, eco-friendly, high ability, non-generation of secondary pollutants, and reusability [11]. Activated carbon is one of the most efficient adsorbents used to treat many

kinds of effluents. Nevertheless, it is relatively expensive, and its regeneration is difficult, which has necessitated the search for alternative adsorbents [12].

Therefore, several industrial and agricultural wastes exhibit considerable potential for the removal of pollutants. Several agricultural by-products are investigated for COD removal from wastewater, including bamboo [13], date pit [14], raw bagasse [15], areca catechu fronds [16], and date palm waste [17]. Among the efficient agro-industrial residues is sugarcane bagasse, which has a high cellulose (45%), hemicellulose (28%), and lignin (18%) content [18]. These materials contain reactive functional groups such as carboxylic and hydroxyl groups with the potential to adsorb organic loads [19]. Several studies have been conducted to investigate the sugarcane bagasse as an adsorbent for different pollutants, including Congo red, Pb(II), phenol from water, and phenol from OMW [20–24].

Although several studies have been conducted using sugarcane bagasse to remove several organic dye contaminants, no investigation has been carried out on the mathematical modeling and statistical optimization of the COD removal from OMW using sugarcane bagasse. Therefore, the present work aimed at the statistical optimization and modeling of key experimental parameters using fractional factorial design and response surface methodology (RSM) to attain optimum COD removal. Moreover, it recognizes the effects of five parameters, such as adsorbent dosage, pH, agitation time, stirring speed, and temperature as well as their interactions on the adsorption process of COD by sugarcane bagasse. Additionally, the adsorption kinetics, equilibrium, and thermodynamics were also studied.

## 2. Materials and Methods

### 2.1. OMW Sample

An OMW solution was obtained from an olive oil mill in Marrakech City, Morocco, during the campaign 2018/2019. This factory operates with the modern two- and three-phase olive oil extraction technology. The OMW used is characterized by a COD of 347.8 g($O_2$)/L, a conductivity of 15.3 ms/cm, a phenolic content of 15.29 g/L and a pH of 4.27. The supplied effluent was immediately kept refrigerated at $-4\,°C$ to prevent any alteration in its physicochemical characteristics.

### 2.2. Sugarcane Bagasse Material

Sugarcane bagasse was collected from a local plant in the region of El Jadida, Morocco. The collected adsorbent was oven-dried for 24 h at 70 °C before being ground with a domestic electrical miller and sieved through a 50 μm screen. The obtained powder was repeatedly washed with distilled water to remove all impurities, and it was then oven dried at 100 °C for 48 h (Figure 1). The dried powder was used in experiments without any further treatment.

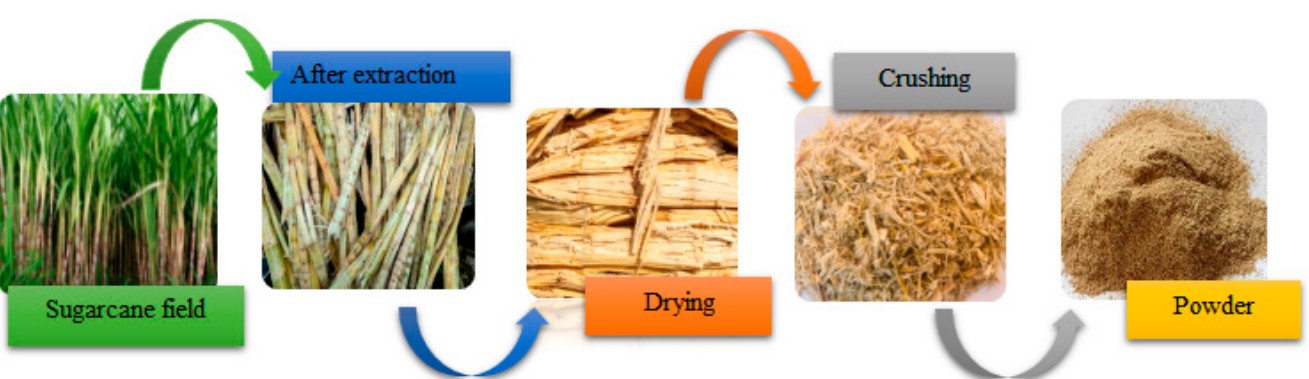

**Figure 1.** Sugarcane bagasse material steps.

The sugarcane bagasse functional groups were characterized by FTIR using Perkin-Elmer 1720-x spectrometer. The sample was mixed with KBr at a mass ratio of 1:100 and finely powdered and pressed to pellets. The infrared spectra were recorded in the range of 4000–500 cm$^{-1}$ with 2 cm$^{-1}$ resolution. Therefore, X-ray fluorescence analysis was employed to investigate the chemical composition of the adsorbent using the "Axion" spectrometer.

### 2.3. Batch Adsorption Experiments

All reagents used in the preparation and the adsorption studies were of analytical grade. The elimination of COD from OMW using sugarcane bagasse was studied in batch mode using shut-top pyrex bottles comprising 100 mL of OMW and an appropriate mass of adsorbent, which were stirred during the desired time in an incubator (Stuart SI50). The remaining COD content was determined from the UV-Vis absorbance characteristic using a Jenway 6320D UV/Vis spectrophotometer. The wavelength of maximum absorption ($\lambda_{max}$) was 620 nm for this measurement. The quantity adsorbed and the removal efficiency of COD were calculated by measuring the solution's concentration before and after adsorption using the following equations:

$$q_e = \frac{(C_o - C_e) \cdot V}{m} \tag{1}$$

$$\%Rem = \frac{(C_o - C_e)}{m} \times 100 \tag{2}$$

where $q_e$ is the amount of COD removed by the adsorbent (mg/g); $C_o$ and $C_e$ are, respectively, the concentrations of COD before and after the batch adsorption study (mg/L); m is the mass of sugarcane bagasse (g); and V is the volume of the solution (L).

The nonlinear optimization method was used to fit equilibrium and kinetics data to the corresponding models. For this study, non-linear regression was applied using 8.0 software for fitting the curve. The best fit of the chosen non-linear models was determined by the use of four well-known error functions, namely the coefficient of determination ($R^2$), adjusted determination coefficient (adj-$R^2$), reduced chi-square (red-$\chi^2$), and Bayesian information criterion (BIC). A model with a higher adj-$R^2$ value, lower red-$\chi^2$ and BIC value indicates a better fitting than the others [25].

The equations of all error functions used are expressed as follows:

$$R^2 = 1 - \frac{\sum (q_{i,exp} - q_{i,model})^2}{\sum (q_{i,exp} - q_{i,exp-mean})^2} \tag{3}$$

$$adj - R^2 = 1 - (1 - R^2) \times \left(\frac{N - 1}{DOF}\right) \tag{4}$$

$$red - \chi^2 = \frac{\sum (q_{i,exp} - q_{i,model})^2}{DOF} \tag{5}$$

$$BIC = N \times \ln\left(\frac{\sum (q_{i,exp} - q_{i,model})^2}{N}\right) + Pln(N) \tag{6}$$

where $q_{i,exp}$ is the experimental adsorbed capacity value; $q_{i,model}$ is the modeled value; $q_{i,exp-mean}$ is an average of $q_{i,exp}$ values used for modelling; N is the number of experimental points; P is the number of model parameters; and DOF is the degrees of freedom.

### 2.4. Experimental Design

Batch experiments based on a $2^{5-1}$ fractional factorial design were conducted randomly to study the influence of the experimental variables on the percentage of COD removal (% Rem). The studied factors which are focused on this research; sugarcane

bagasse dose ($X_1$), solution pH ($X_2$), contact time ($X_3$), stirring speed ($X_4$) and temperature ($X_5$). Table 1 shows the five experimental variables and their chosen levels. After performing batch experiments for the optimization process, the regression analysis was conducted to attain the study's statistical parameters with 95% confidence intervals using the Minitab 18 statistical software. The variance of the regression equation (mathematical relation between dependent and independent variables) is the most important analysis by the ANOVA method to find the desired function of COD adsorption. RSM is used as a sequential process to show the relation between the studied independent factors and the response to determine the set of optimal experimental parameters.

**Table 1.** Process factors and their levels.

| Factors | Variable | Unit | Levels | |
|---|---|---|---|---|
| | | | Low ($-$) | High ($+$) |
| $X_1$ | Adsorbent dose | g/L | 10 | 60 |
| $X_2$ | pH | - | 2 | 12 |
| $X_3$ | Contact time | h | 1 | 24 |
| $X_4$ | Stirring speed | rpm | 80 | 300 |
| $X_5$ | Temperature | °C | 25 | 60 |

## 3. Results and Discussion

### 3.1. Analysis of Factorial Design

$2^{5-1}$ Fractional factorial design was adopted using Minitab 18 to optimize the influence of the investigated parameters; sugarcane bagasse dose ($X_1$), solution pH ($X_2$), contact time ($X_3$), stirring speed ($X_4$) and temperature ($X_5$) on the elimination of COD from OMW. This design yields in 16 experiments with all possible combinations of $X_1$, $X_2$, $X_3$, $X_4$ and $X_5$. COD removal efficiency (Y) was measured for each of these experiments as shown in Table 2. The response obtained was correlated using the second-order polynomial model, expressed by Equation (7):

$$Y = b_0 + b_1X_1 + b_2X_2 + b_3X_3 + b_4X_4 + b_5X_5 + b_{12}X_1X_2 + b_{13}X_1X_3 + b_{14}X_1X_4 + b_{15}X_1X_5 + b_{23}X_2X + b_{24}X_2X_4 + b_{34}X_3X_4 + b_{25}X_2X_5 + b_{35}X_3X_5 + b_{45}X_4X_5 \tag{7}$$

where Y is the COD removal efficiency response, $b_0$ is a constant, bi correspond to linear coefficient of Xi, and bij is the interaction coefficient.

**Table 2.** Matrix design with experimental and predicted COD removal values.

| N° | $X_1$ | $X_2$ | $X_3$ | $X_4$ | $X_5$ | Y (Experimental) | Y (Predicted) |
|---|---|---|---|---|---|---|---|
| 1 | $-1$ | $-1$ | $-1$ | $-1$ | $+1$ | 36.39 | 37.46 |
| 2 | 1 | $-1$ | $-1$ | $-1$ | $-1$ | 41.26 | 43.47 |
| 3 | $-1$ | 1 | $-1$ | $-1$ | $-1$ | 54.27 | 51.69 |
| 4 | 1 | 1 | $-1$ | $-1$ | $+1$ | 44.69 | 43.75 |
| 5 | $-1$ | $-1$ | 1 | $-1$ | $-1$ | 35.24 | 35.54 |
| 6 | 1 | $-1$ | 1 | $-1$ | $+1$ | 42.40 | 43.24 |
| 7 | $-1$ | 1 | 1 | $-1$ | $+1$ | 35.53 | 30.66 |
| 8 | 1 | 1 | 1 | $-1$ | $-1$ | 41.23 | 40.85 |
| 9 | $-1$ | $-1$ | $-1$ | 1 | $-1$ | 40.71 | 40.53 |
| 10 | 1 | $-1$ | $-1$ | 1 | $+1$ | 42.12 | 43.90 |
| 11 | $-1$ | 1 | $-1$ | 1 | $+1$ | 33.73 | 32.73 |
| 12 | 1 | 1 | $-1$ | 1 | $-1$ | 39.82 | 39.12 |
| 13 | $-1$ | $-1$ | 1 | 1 | $+1$ | 33.23 | 29.86 |
| 14 | 1 | $-1$ | 1 | 1 | $-1$ | 31.12 | 34.75 |
| 15 | $-1$ | 1 | 1 | 1 | $-1$ | 39.25 | 38.14 |
| 16 | 1 | 1 | 1 | 1 | $+1$ | 42.12 | 41.79 |

The statistical calculations and regression analysis were conducted to fit the response function with the experimental data. The regression coefficient values obtained are given in the final regression equation, after putting the values of all coefficients, as follows (Equation (8)):

$$Y\,(\%) = 37.85 - 2.183X_1 + 0.39X_2 - 0.562X_3 + 0.05958X_4 + 0.2182X_5 - 0.01712X_1X_3 - 0.0529X_1X_3 + 0.102X_1X_4 - 0.0528X_1X_5 - 0.05223X_2X_3 - 0.01976X_2X_4 - 0.01153X_2X_5 - 0.003092X_3X_4 - 0.00436X_3X_5 - 0.000623X_4X_5 \tag{8}$$

This equation expresses the COD removal efficiency as a function of investigated experimental factors and enables fixing experimental conditions for each targeted COD removal efficiency. From Table 2, the values predicted obtained by the relation (8) are compared with those of experimental results, which show a good agreement between the two sets of values. In Equation (8), the positive signs of the coefficients for $X_2$ and $X_4$ factors and the $X_1X_4$ interaction imply their positive effect on the response, while the negative signs of the coefficients for $X_1$, $X_3$, and $X_5$ factors as well as the $X_1X_2$, $X_1X_3$, $X_1X_5$, $X_2X_3$, $X_2X_5$, $X_3X_4$, $X_3X_5$, and $X_4X_5$ interactions represent the negative effect on the response. Based on the equation, the most influential factor for response was the sugarcane bagasse dose with a coefficient value of 2.183. The sign (-) means that each one-point decrease will have an effect of 2.183 on the COD removal efficiency value.

To check the quality of the model fitting, an ANOVA analysis was performed using the F-value and $p$-value (Table 3). Significant effects of the model terms were identified based on the $p$-values less than 0.05. Since the F-value for the 95% confidence interval, one degree of freedom, and 16 factorial runs, was equal to 4.54, all terms of the model with a F-value greater than 4.54 are considered statistically significant. The COD removal model was determined to be greatly significant from Fisher's test (F-value of 22.24) and a smaller $p$-value (<0.0001). Besides that, the significance of the model terms can be determined through the Pareto chart (Figure 2). The Student's $t$-test was performed to evaluate the significance of the regression coefficients. For the 95% confidence interval and one degree of freedom, it was observed that the t-value was equal to 2.12. Student's t-test values for model terms are displayed in horizontal columns. All term effects are significant if their absolute values exceed the vertical line (2.12). According to the obtained F value, $p$-value and Pareto chart, it seems that the main effects $X_1$, $X_2$, $X_3$, $X_4$ and $X_5$ factors as well as the $X_2X_3$, $X_1X_4$, $X_3X_5$, $X_1X_2$, $X_3X_4$, $X_4X_5$ and $X_1X_3$ interactions are statistically significant. In this way, the COD adsorption by sugarcane bagasse could be expressed using the following equation (Equation (9)):

$$Y\,(\%) = 21.5 - 0.2439\,X_1 + 1.494\,X_2 + 1.387\,X_3 - 0.0451\,X_4 + 0.4404\,X_5 - 0.01712\,X_1X_2 - 0.00497\,X_1X_3 + 0.001892\,X_1X_4 - 0.09223\,X_2X_3 - 0.001131\,X_3X_4 - 0.01268\,X_3X_5 - 0.000743\,X_4X_5 \tag{9}$$

To graphically verify the validity of the regression model obtained for COD removal, four validation indicator plots were created by plotting the differences between the predicted (model) and the observed (experimental) values (Figure 3).

The normal probability plot (Figure 3a) reveals that all points are reasonably near a straight line, suggesting that the predicted values of COD removal and the actual experimental data were in agreement, evidencing the normal distribution of the data and the validity of the regression model. The graphic plot of the residuals (Figure 3b) displays the predicted and observed values. The residual (vertical axis) is the difference between the experimental and the fitted values [26]. As shown in Figure 3b, the experimental points are reasonably aligned around zero, suggesting normal distribution. The histogram (Figure 3c) shows a random distribution of values with no noticeable shape or trend across all 16 runs conducted. The residuals versus the observation orders (Figure 3d) show that the residuals seem to be randomly scattered around zero.

**Table 3.** Analysis of ANOVA for COD removal. The underline shows the *p*-values < 0.05.

| Source | DF | Adj SS | Adj MS | F-Value | *p*-Value |
|---|---|---|---|---|---|
| Model | 16 | 3214.58 | 200.912 | 22.24 | 0.000 |
| Linear | 5 | 881.04 | 176.207 | 19.51 | 0.000 |
| $X_1$ | 1 | 224.88 | 224.876 | 24.90 | 0.000 |
| $X_2$ | 1 | 192.57 | 192.567 | 21.32 | 0.000 |
| $X_3$ | 1 | 146.82 | 146.823 | 16.25 | 0.001 |
| $X_4$ | 1 | 260.52 | 260.525 | 28.84 | 0.000 |
| $X_5$ | 1 | 56.25 | 56.245 | 6.23 | 0.025 |
| Interactions | 10 | 2327.91 | 232.791 | 25.77 | 0.000 |
| $X_1 \times X_2$ | 1 | 146.54 | 146.538 | 16.22 | 0.001 |
| $X_1 \times X_3$ | 1 | 65.35 | 65.350 | 7.23 | 0.017 |
| $X_1 \times X_4$ | 1 | 865.84 | 865.839 | 95.86 | 0.000 |
| $X_1 \times X_5$ | 1 | 5.31 | 5.314 | 0.59 | 0.455 |
| $X_2 \times X_3$ | 1 | 900.00 | 899.998 | 99.64 | 0.000 |
| $X_2 \times X_4$ | 1 | 0.34 | 0.343 | 0.04 | 0.848 |
| $X_2 \times X_5$ | 1 | 5.26 | 5.260 | 0.58 | 0.457 |
| $X_3 \times X_4$ | 1 | 65.50 | 65.502 | 7.25 | 0.017 |
| $X_3 \times X_5$ | 1 | 208.31 | 208.306 | 23.06 | 0.000 |
| $X_4 \times X_5$ | 1 | 65.46 | 65.464 | 7.25 | 0.017 |
| Error | 15 | 135.49 | 9.033 | | |
| Total sum of squares | 31 | 3350.07 | | | |

| S square | $R^2$ | $R^2$ (ajust) | $R^2$ (prev) |
|---|---|---|---|
| 6.2538 | 99.98% | 87.54% | 62.23% |

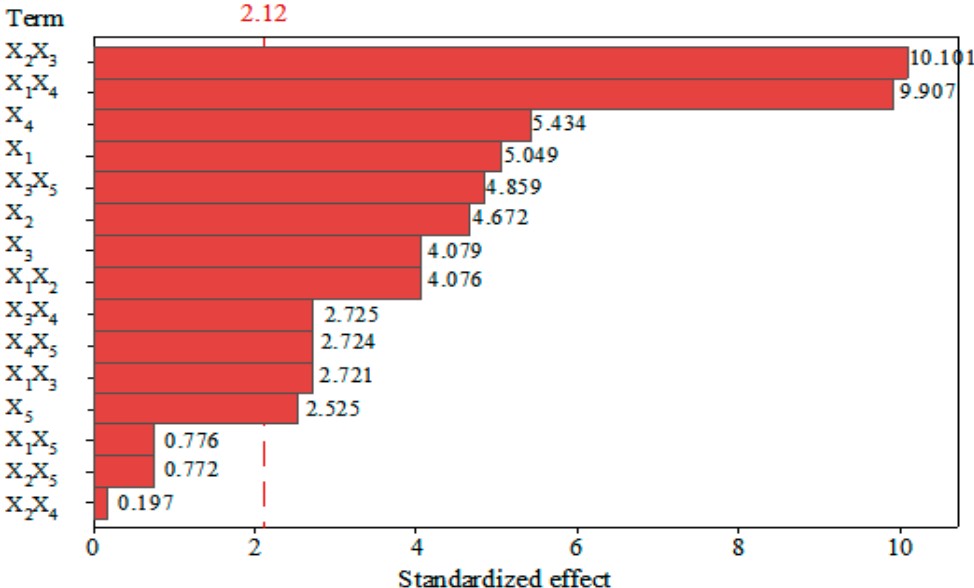

**Figure 2.** Pareto chart for standardized effects.

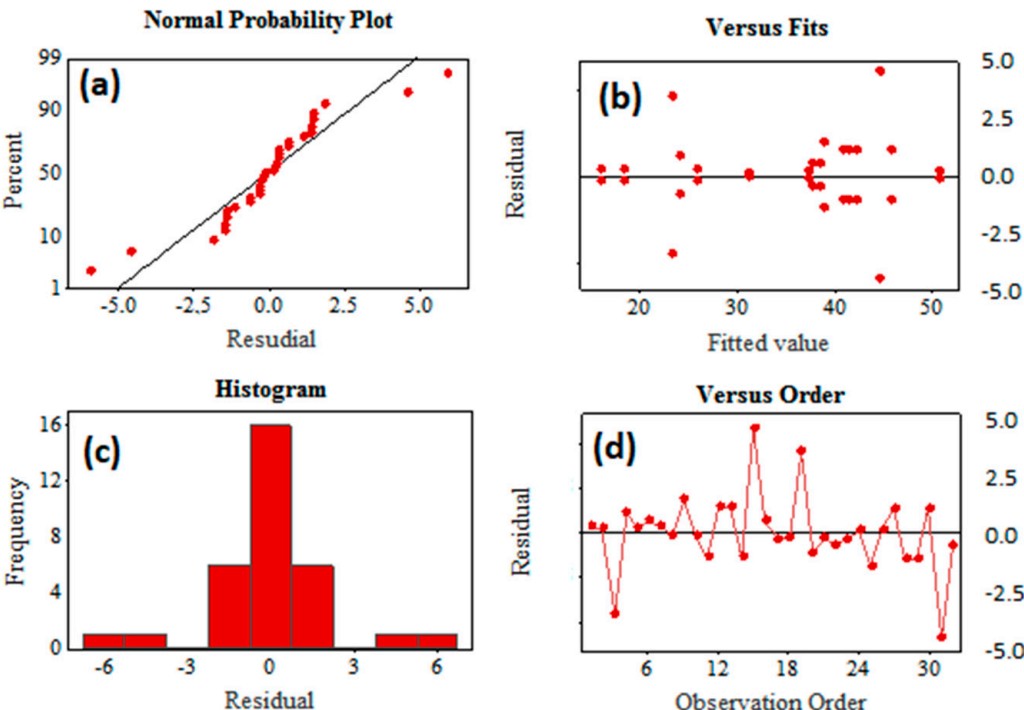

**Figure 3.** Validation indicator plots of the model for the experiments (**a**) Normal probability plot, (**b**) Versus fits (**c**) Histogram and (**d**) Versus order.

*3.2. Response Surface Analysis*

The significant interaction effect between each two input parameters on the % removal was presented by the 3D response surface and the contour of the plots (Figure 4A–G). The interaction between the solution pH ($X_2$) and contact time ($X_3$) is shown in Figure 4A. The graph demonstrated that better COD removal is achieved at acidic pH and long contact time. Further, the % removal decreased from 39 to 30% with increasing pH. This can be explained by the impact of pH on the ionic configuration of the functional groups presented in the sorbent. Previous pH drift tests indicated that the $pH_{pzc}$ (point of zero charge) of sugarcane bagasse is equal to 5.0, which expresses that the surface of the bagasse is positively charged at a pH below 5 and negatively charged at a pH above 5 [21,27]. As a consequence, the electrostatic interaction between the adsorbent surface and the organic matter could be enhanced, which resulted in high adsorption of COD [28]. The interaction effect between the adsorbent dose ($X_1$) and agitation speed ($X_4$) shown in Figure 4B reveals that there will be a good COD removal equal to 44% when the adsorbent dose is 10 g/L and also at the stirring speed of 80 rpm. Nevertheless, an adsorbent dose greater than 10 g/L and an agitation speed above 80 rpm result in a substantial decrease in % removal. This phenomenon is due to the aggregation and glomeration of the sorbent particles and the reduction in the total surface area of the adsorbent [29].

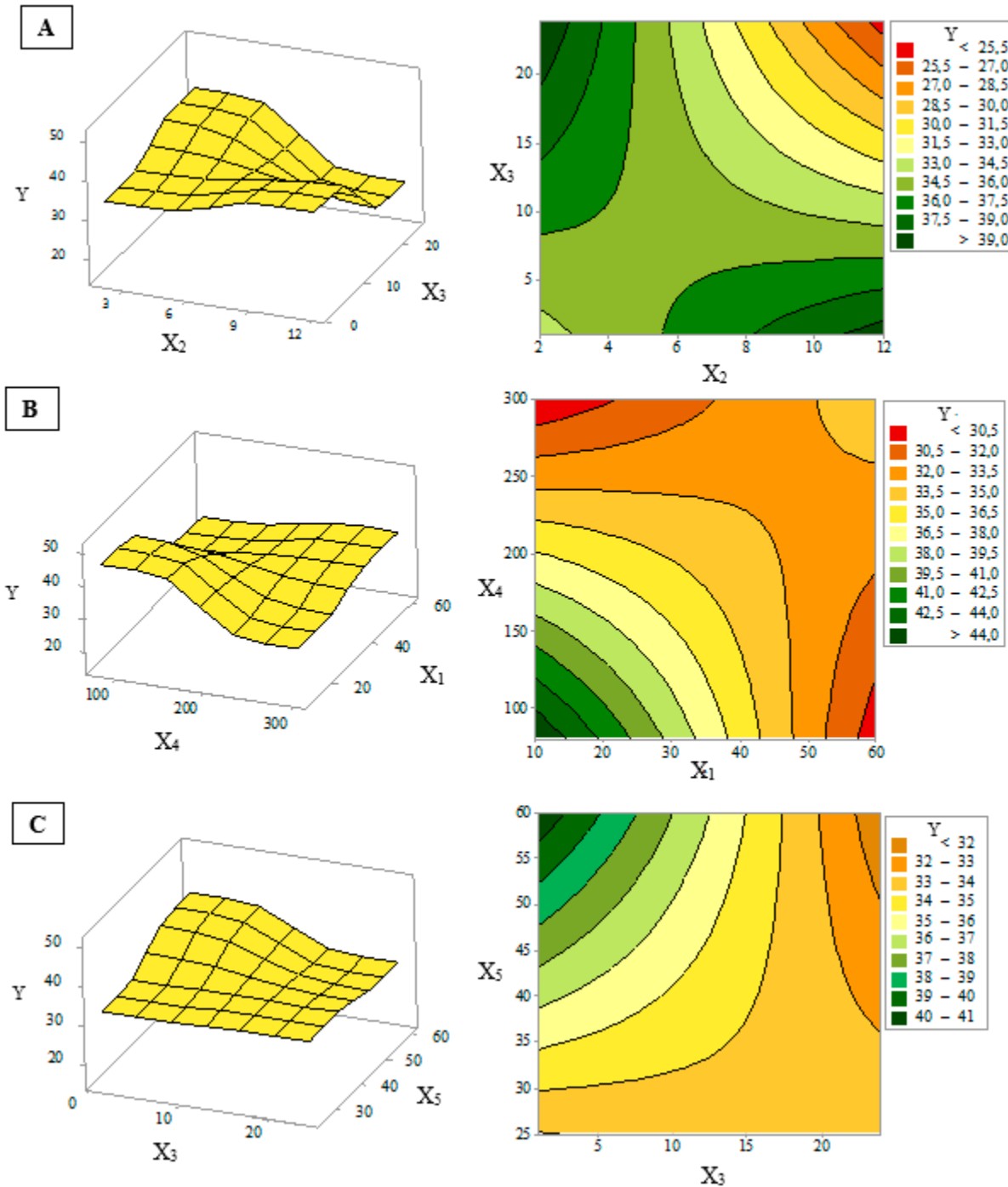

**Figure 4.** *Cont.*

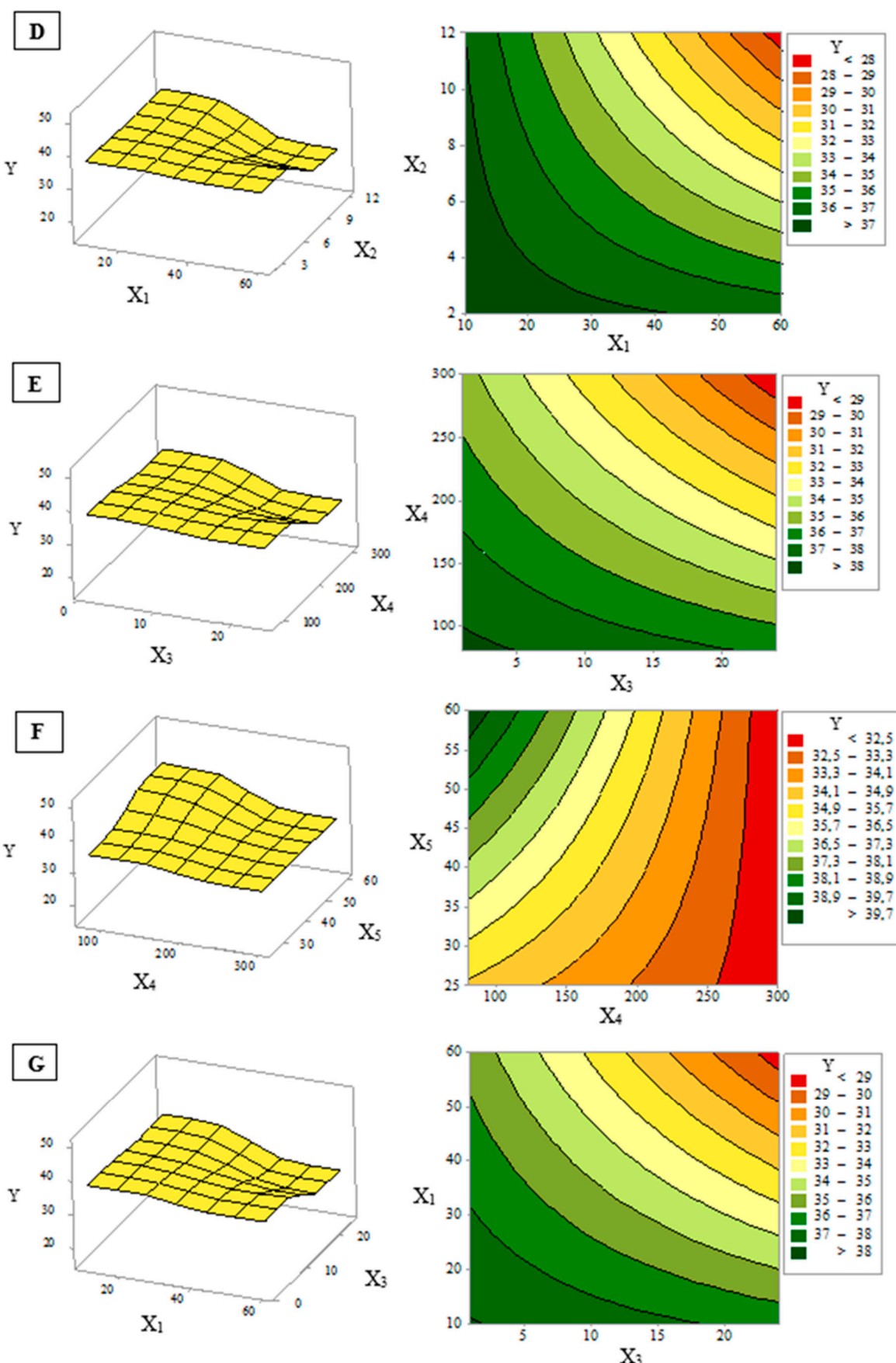

**Figure 4.** (**A–G**): Response surface and contour plots for the % Rem.

Figure 4C describes the interaction between the temperature ($X_5$) and reaction time ($X_3$) on the % removal. The % removal increases with an increase in temperature, whereas it decreases with increasing contact time. There are more active surface sites when the temperature increases because the adsorbent swells more as well. This result suggests that the COD adsorption is an endothermic process [30]. As shown in Figure 4D, the interaction between the pH ($X_2$) and adsorbent dose ($X_1$) has a negative effect on % removal. An increase in either of these two parameters reduces the COD removal efficiency. The possible reason for such observations is that the changes in pH could lead to changes in the properties of the surface of adsorbent [31], and the agglomeration of the adsorbate particles occurs with an increase in the adsorbent dose. In Figure 4E, the % removal decreased with increasing contact time ($X_3$) and stirring speed ($X_4$). The % removal was very rapid in the beginning stages of contact time, reaching about 38%. This result was related to the availability of more active sites on the adsorbent and the fact that the gradient of concentration between the adsorbate molecules in the solution and the adsorbate molecules on the adsorbent is high, which improves COD diffusion to the adsorbent surface [32,33]. After 60 min, the removal decreases over time due to adsorption site saturation, and the adsorbate molecules may bind poorly to the active receptors on the adsorbent [34,35].

Furthermore, the obtained results represented in Figure 4F show that the optimum of % removal was found at a temperature of 60 °C and agitation speed of 80 rpm, giving 39% of the recovery. This tendency may be due to the breakdown of the expanding chain and flocs. However, the higher stirring speed can encourage the process of agglomeration [36]. Additionally, the % removal was decreased by increasing the adsorbent dosage ($X_1$) and contact time ($X_3$) (Figure 4G). The % removal is initially higher and more rapid, reaching up to more than 38% within the first 60 min and 10 g/L, which could be attributed to the greater availability of adsorption sites to bind COD and a greater adsorbent active sites/COD ratio [37]. Subsequently, it decreased slightly with the prolonging of the contact time and the increase in adsorbent dose, which could be caused by an aggregation of the adsorbent and the reduction of the surface area available to COD [20].

### 3.3. Optimization Process

The response optimization study used to determine the combination of optimal values of the factors for maximum COD removal is shown in Figure 5. The validation of the optimal conditions is regarded as the final step in the modeling approach to examine the precision and robustness of the investigated model. The optimization requests to determine the desired pH, contact time, temperature, stirring speed, and adsorbent concentration to attain significant desirability. The model optimization shows that the efficiency of COD removal increases with an increasing pH, contact time and temperature, while decreasing with an increasing adsorbent dose and stirring speed. The optimum conditions indicate that the experimental modeling yields a desirability close to 1 with 55.07% COD removal efficiency, in the subsequent conditions: 10 g/L doses of sugarcane bagasse, 1 h of contact time, 80 rpm of stirring speed, 60 °C of temperature, and pH 12 of the OMW solution.

### 3.4. Characterization of Sugarcane Bagasse

Sugarcane bagasse characterization is an important analysis for understanding the behavior or the mechanism of COD removal on its surface. X-ray fluorescence elemental analysis was used to identify and quantify the amount of the elements contained in sugarcane bagasse in order to ultimately determine its elemental composition. The mineral composition of sugarcane bagasse is presented in Table 4. The results demonstrate that sugarcane bagasse contains a high amount of silicon dioxide ($SiO_2$), as high as 62.23%, and low amounts of alkaline oxide, $Al_2O_3$ and $P_2O_5$.

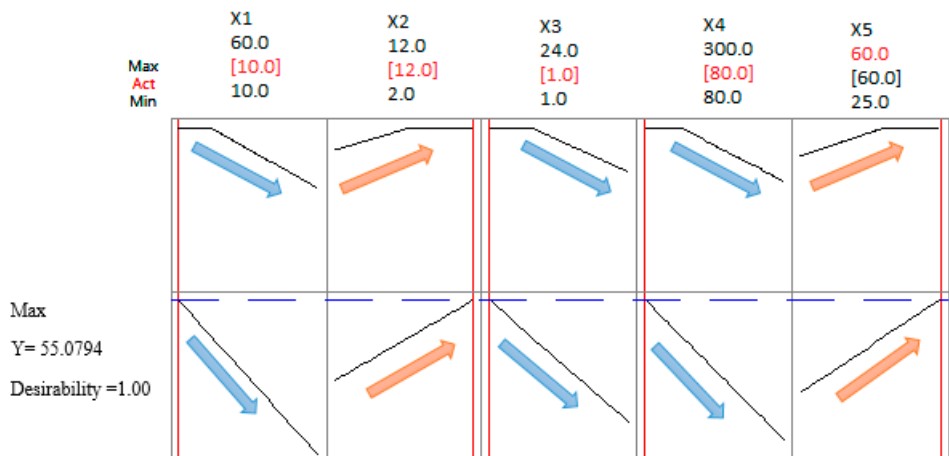

**Figure 5.** Optimization of the model obtained by desirability function.

**Table 4.** X-ray fluorescence analysis of sugarcane bagasse.

| Composition | Concentration (%) |
| --- | --- |
| $SiO_2$ | 62.23 |
| $SO_3$ | 24.00 |
| CaO | 11.50 |
| MgO | 0.68 |
| $Na_2O$ | 0.60 |
| $K_2O$ | 0.51 |
| $Al_2O_3$ | 0.28 |
| $P_2O_5$ | 0.1 |
| $Fe_2O_3$ | 0.10 |

Therefore, the FTIR was utilized to investigate the functional groups of the sugarcane bagasse (Figure 6). The bands at 3330 and 2890 cm$^{-1}$ indicate the presence of the O-H functional group and C–H stretching, respectively [20,38]. The peak at 1632 cm$^{-1}$ is assigned to C=O vibrations in hemicellulose [39]. These groups are thought to play a very important role in the process of adsorption [40]. The bands that appeared at wave numbers between 1471 cm$^{-1}$ and 1366 cm$^{-1}$ are related to C=C–H indicating several bands in cellulose and xylose [41]. The bands appeared at 1241 cm$^{-1}$ and 1029 cm$^{-1}$ can be attributed to the CH=CH stretching of lignin [39] and C-O stretching in cellulose and hemicellulose [42], respectively. The bands appeared at 640 and 593 cm$^{-1}$ in the FTIR spectra are attributed to the vibration of O-H groups out of the plane deformation [43].

*3.5. Adsorption Isotherms*

The specific relationship established between the COD remaining in solution (Ce) and the COD adsorbed by sugarcane bagasse ($q_e$) at equilibrium was analyzed by the models of Langmuir and Freundlich. The investigated adsorption isotherms for the COD adsorption on sugarcane bagasse are presented in Figure 7. Table 5 shows the calculated parameters for each of the sorption isotherm models obtained from non-linear regression forms. The best fit of the experimental data was analyzed based on $R^2$, adj-$R^2$, red-$\chi^2$ and BIC. The obtained results displayed that the COD adsorption fitted well with the Langmuir isotherm model with the highest adj-$R^2$ of 0.994 and the lowest red-$\chi^2$ and BIC values of 78.33 and 25.95, respectively. This result indicates that the adsorption process occurs on a homogeneous surface, and all adsorption sites are identical and energetically equivalent [44]. The maximum adsorption capacity ($q_{max}$) of COD onto sugarcane bagasse was determined to be 331.92 mg/g from the Langmuir model.

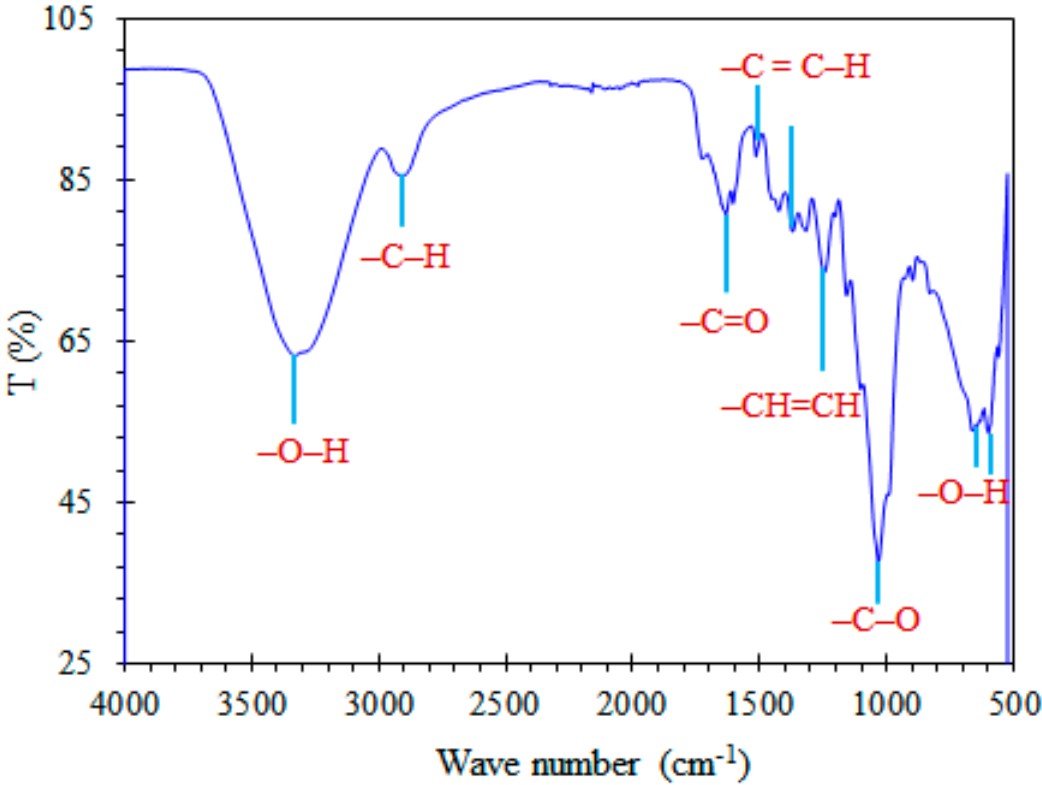

**Figure 6.** The FTIR spectra of sugarcane bagasse.

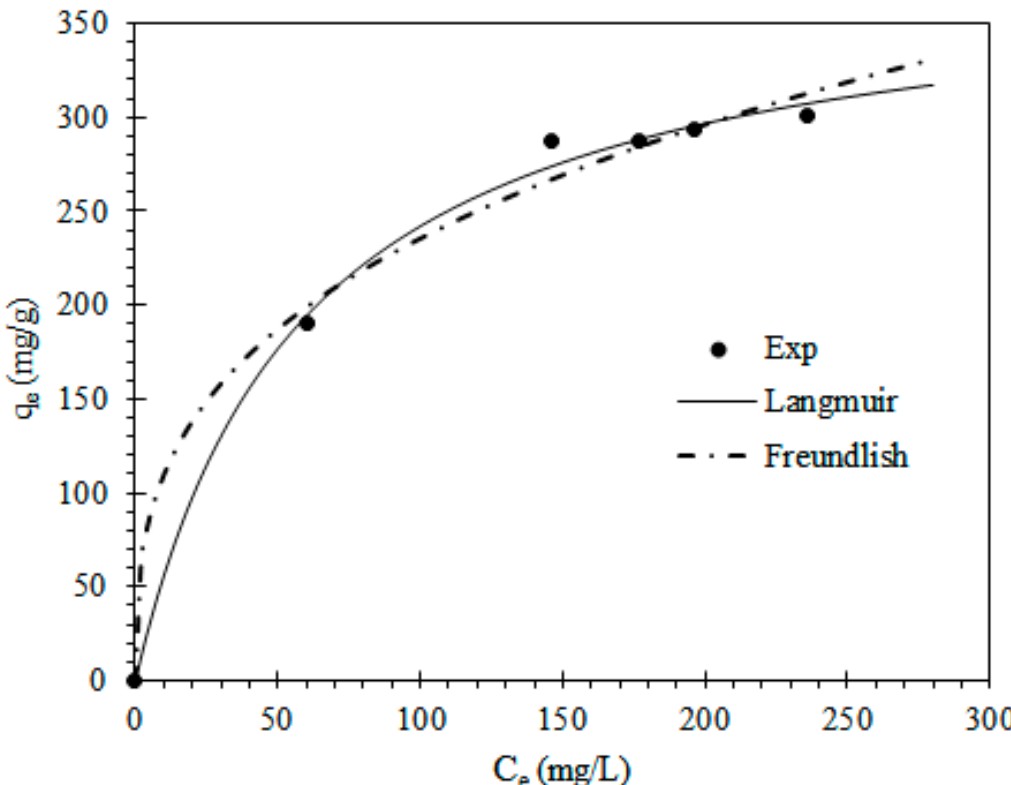

**Figure 7.** Isotherm plots for COD removal by sugarcane bagasse.

**Table 5.** Parameters and error function data for sorption isotherm models obtained from non-linear regression forms.

| Models | Equation [a] | Parameters | Values |
|---|---|---|---|
| Langmuir | $q_e = \frac{q_{max}K_LC_e}{(1+K_LC_e)}$ | qm (mg/g) | 331.92 |
| | | $K_L$ | 0.019 |
| | | $R^2$ | 0.996 |
| | | Adj-$R^2$ | 0.994 |
| | | Red-$\chi^2$ | 78.33 |
| | | BIC | 25.59 |
| Freundlich | $q_e = K_FC_e^{1/n}$ | 1/n | 0.31 |
| | | $K_F$ | 51.13 |
| | | $R^2$ | 0.991 |
| | | Adj-$R^2$ | 0.985 |
| | | $\chi^2$ | 208.11 |
| | | BIC | 31.45 |

Note: [a] $q_e$ (mg/g): mass of adsorbed molecule per unit mass of sugarcane bagasse, $C_e$ (mg/L): concentration of no-adsorbed molecules, $q_m$ and $K_L$ are constants of the Langmuir model, and $K_F$ and n are constants of Freundlich isotherm model.

The adsorption properties of sugarcane bagasse are compared with other adsorbents used for COD sorption reported in the literature, which are given in Table 6. The table indicates that the $q_{max}$ value obtained in the present study is higher than that of other materials from previous studies. The $q_{max}$ value found in this study reveals a very good adsorption capacity of the sugarcane bagasse, which falls as a promising adsorbent.

**Table 6.** Comparison of the $q_{max}$ of COD removal by various adsorbents.

| Adsorbent Types | $q_{max}$ (mg/g) | Ref. |
|---|---|---|
| Sugarcane bagasse | 331.92 | This study |
| Date-pit activated carbon | 252.81 | [14] |
| Activated carbon | 41.3 | [17] |
| Raw bagasse | 77.95 | [15] |
| Bamboo-based activated carbon | 24.39 | [45] |
| Areca Nut Husk | 64.94 | [46] |

*3.6. Kinetic Studies*

Kinetic modeling was undertaken to determine the rate of COD adsorption on the sugarcane bagasse and examine the controlling mechanisms of the adsorption process. The kinetic studies were carried out from 0 to 24 h, and the recorded data were studied with the pseudo-first-order and pseudo-second-order kinetic models. The rate constants values were valued from the non-linear plots and shown in Figure 8 and are also summarized in Table 7. The table indicated that the $R^2$ value obtained from the pseudo-first order kinetic equation was found to be higher than that of pseudo-second order. In addition, the calculated $q_e$ value (337.45 mg/g) for the pseudo-first-order model is closer to the experimental value (326.29 mg/g) compared to the $q_e$ value (405.9 mg/g) calculated for the pseudo-second-order model. The pseudo-first-order model also presented the highest adj-$R^2$ (0.967), lowest red-$\chi^2$ (467.41) and BIC (43.01) values. This result advises that the experimental data demonstrated the best fit to the pseudo-first order model, and its applicability also indicates that a physical process might control the sorption process.

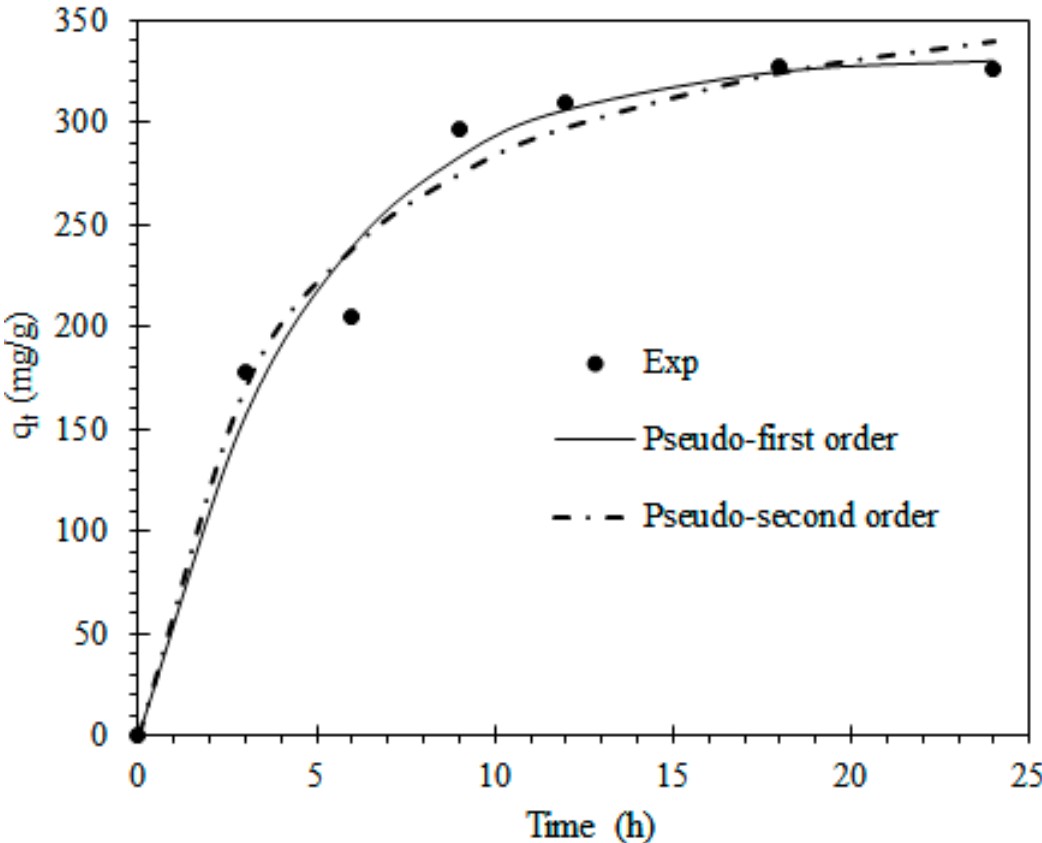

**Figure 8.** Kinetic plots for COD removal by bagasse.

**Table 7.** Parameters and error functions data for kinetic models studied obtained from non-linear regression forms.

| Model | Equation [b] | Parameters | Value |
|---|---|---|---|
| Pseudo-first-order | $q_t = q_e \left(1 - e^{-k_1 t}\right)$ | $q_{e.exp}$ (mg/g) | 326.92 |
| | | $q_{ecal}$ (mg/g) | 337.45 |
| | | $k_1$ | 0.20 |
| | | $R_1^2$ | 0.978 |
| | | Adj-$R^2$ | 0.967 |
| | | $\chi^2$ | 467.41 |
| | | BIC | 43.01 |
| Pseudo-second-order | $q_t = \dfrac{K_2 q_e^2 t}{(1 + K_2 q_e t)}$ | $q_{ecal}$ (mg/g) | 405.9 |
| | | $k^2$ | 0.0006 |
| | | $R_2^2$ | 0.976 |
| | | Adj-$R^2$ | 0.964 |
| | | $\chi^2$ | 503.52 |
| | | BIC | 43.52 |

Note: [b] $q_e$ and $q_t$ (mg/g) are the amounts of COD removed at equilibrium and at time t, respectively; $k_1$ and $k_2$ are the adsorption rate constants.

## 4. Conclusions

In this work, the sugarcane bagasse prepared was utilized as an adsorbent for the elimination of COD from OMW by the adsorption. Sugarcane bagasse characterization was performed by FTIR and X-ray fluorescence. The $2^{1-5}$ fractional design associated to response surface methodology was successfully applied to optimize the effects of the operating variables of COD removal from OMW using the prepared adsorbent. The optimal conditions were found to be pH 12, the adsorbent dose of 10 g/L, a stirring speed of 80 rpm, a contact time of 1 h, and a temperature of 60 °C with a percentage of removal of 55.07%

and a desirability close to 1. The experimental data are well correlated by the Langmuir isotherm model with an $R^2$ of 0.996, and the maximum sorption capacity of 331.92 mg/g under optimal conditions. The kinetic data of the COD removal process were properly fitted with the pseudo-first-order model instead of pseudo-second-order model.

**Author Contributions:** F.E. carried out the experiments and prepared the draft manuscript. S.E.: and W.B. interpreted and discussed the results, and calculated the statistical parameters. N.B. (Nadia Beniich) analyzed and checked the statistical results. N.B. (Noureddine Barka) verified the analytical methods. C.E.A. and E.L. aided in interpreting the results and worked on the manuscript. M.A. corrected and wrote the final version of the manuscript. All authors have read and agreed to the published version of the manuscript.

**Funding:** This research received no external funding.

**Data Availability Statement:** The data presented in this study are available on request from the corresponding author.

**Conflicts of Interest:** The authors declare no conflict of interest.

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
