# Peer review of "Factorial Design Statistical Analysis and Optimization of the Adsorptive Removal of COD from Olive Mill Wastewater Using Sugarcane Bagasse as a Low-Cost Adsorbent"

_water, doi:10.3390/w15081630_

Round 1

Reviewer 1 Report

Comments to the Author

In general, this article has met all criteria required by the journal. Methodology, experimental design, explanation, and discussion of the results are presented clearly enough and not confuse the reader. The results can give a positive value in the research area of removal of COD from waste water. Major revision is required before publication. Detailed comments are listed as follows:

1. Will the obtained bagasse powder ignite when dried at 100℃?Or is there a safety hazard? (Line 74-75)

2. Does the sugarcane bagasse have to be negatively charged? Perhaps there is any literature or data to support it. (Line 188-190)

3. A 3D response surface diagram can be added in Figure 4.

4. It would be better to add SEM of sugarcane bagasse.

5. What is the function of the groups that correspond to these absorption peaks? (Line 257-260)

6. There are some grammatical errors, please check whole the manuscript for them.

Author Response

Review Report Form 1

Comments and Suggestions for Authors

Point 1: Will the obtained bagasse powder ignite when dried at 100℃?Or is there a safety hazard? (Line 74-75).

Response 1: According to Guillou et al., 2020, evaporation of the water in the bagasse can be realized at 50°C. To ensure that all the water contained in the bagasse is evaporated, we raise the temperature to 70°C during our study. The behavior of bagasse at 100°C remains unknown as the experiment was not conducted at this temperature.

P., Guillou, O.M., Marc, L., Adelard, D., Madyira, E., Akinlabi, et al.. Séchage

de copeaux de bois et de bagasse : simulation numérique et comparaison expérimentale. Congrès Annuel de la Société Française de Thermique 2020, Jun 2020, Belfort, France. ff10.25855/SFT2020- 018ff. ffhal-03270974f

Point 2: Does the sugarcane bagasse have to be negatively charged? Perhaps there is any literature or data to support it. (Line 188-190).

Response 2: Previous pH drift tests indicated that the pHpzc (point of zero charge) of sugarcane bagasse is equal to 5.0, which expresses that the surface of the bagasse is positively charged at a pH below 5 and negatively charged at a pH above 5 [24,25]. As a consequence, the electrostatic interaction between the adsorbent surface and the organic matter could be enhanced which resulted in high adsorption of COD.

This sentence was added in the revised manuscript with some adequate citations to improve this analysis [Line 217-22].

[24] A. Moubarik, N. Grimi. Valorization of olive stone and sugar cane bagasse by-products as biosorbents for the removal of cadmium from aqueous solution. Food Research International. 73 (2015) 169–175.

[25] Z. Zhang, L. Moghaddam, I. M. O’Hara, W.O.S. Doherty. Congo Red adsorption by ball-milled sugarcane bagasse. Chemical Engineering Journal 178 (2011) 122– 128.

Point 4: A 3D response surface diagram can be added in Figure 4.

Response 3: 3D surface plots are presented in the left of figure 4. The used software allows only the reported type of plot. 

Point 5: It would be better to add SEM of sugarcane bagasse.

Response 5: We agree with you, the SEM is a very important technique to compare between bagasse morphology before and after adsorption. However, the SEM material is not available in our establishment.

Point 6: What is the function of the groups that correspond to these absorption peaks? (Line 257-260).

Response 6: The bands that appear at wave numbers between 1603 cm-1 and 1385 cm-1 are related to C=C associated with the aromatic ring of lignin.

Point 7: There are some grammatical errors, please check whole the manuscript for them.

Responses 7: The manuscript has been revised to incorporate improvements.

Reviewer 2 Report

The analyzes and research presented in the article are interesting and worthy of interest by a larger group of readers, but in my opinion the article needs to be corrected and put in order. I have the impression that it touches on several topics without explaining the purpose of these analyses. The article requires primarily editorial corrections. in this version it gives the impression of an unfinished article, e.g.:
1. Please explain, provide the full name of the OMW designation in the title of the article
2. Since the authors mention that the cost of activated carbon is high compared to other materials, this should be supported by an economic analysis.
3. In my opinion, the chapter: Materials and methods needs to be tidied up. In the version submitted for review, descriptions of materials and methods are disorganized and inconsistent.
4. In chapter 3 we have several (5) subchapters 2.2. This makes it difficult to read and interpret the text.
I think the authors should edit the text again. In my opinion, this version needs some improvements.
Thank you for considering my opinion. I encourage the authors to continue working on improving the manuscript.

Author Response

Review Report Form 2

Comments and Suggestions for Authors

Point 1: Please explain, provide the full name of the OMW designation in the title of the article.

Response 1: The full name of the OMW designation is provided in the title of the article.

Point 2: Since the authors mention that the cost of activated carbon is high compared to other materials, this should be supported by an economic analysis.

Response 2: It is well documented that activated carbon has high cost compared to natural biosorbents due to the cost of activation and recovery. Mohan and Pittman, 2006 reported that activated carbon is a costly material with a price of approximately US$ 1.10 per kilogram, while, the bagasse is among the low-cost adsorbents used for wastewater treatment. The paper of Mohan and Pittman provides an estimation cost of activated carbons and other adsorbents. The Mohan and Pittman paper is added in our manuscript [Line 51]. 

Point 3: In my opinion, the chapter: Materials and methods needs to be tidied up. In the version submitted for review, descriptions of materials and methods are disorganized and inconsistent.

Response 3: The material and method part has been reorganized. The chapter 2 has been subdivided into four subchapters as outlined in the revised version.

Subchapter 2.1 describes the OMW origin and characterization.

Subchapter 2.2 describes sugarcane bagasse material.

Subchapter 2.3 describes the adsorption process.

Subchapter 2.4 describes the experimental design.

Point 4: In chapter 3 we have several (5) subchapters 2.2. This makes it difficult to read and interpret the text.

Response 4: The chapter 3 has been reorganized. The number of subchapters has been reduced and the interpretation has been reformed.

Point 5: I think the authors should edit the text again. In my opinion, this version needs some improvements. Thank you for considering my opinion. I encourage the authors to continue working on improving the manuscript.

Response 5: The manuscript has been revised to incorporate improvements.

Round 2

Reviewer 1 Report

OK

Author Response

Response to academic Editor’s comments

Review Report Form 1

Comments and Suggestions for Authors

Point 1: The resolution of most figures are not high, please increase them (resolution should be higher than 600 ppi).

Response 1: The resolution of all figure is modified.

Point 2: The information in Table 5. X-ray fluorescence analysis of sugarcane bagasse should be revised. The X-ray fluorescence analysis cannot provide the information on C, O, H, Si...., please recheck and correct it.

Response 2: Thank you for your remark. The information in Table 5 is revised and corrected.

Point 3: For FTIR, the authors should (1) change the word “peak” (i.e., “The peaks at 3330 and 2890 cm-1”) to “band”, revise (2) the x-axis from 4000 to 510 cm-1 (rather than 510 to 400 cm-1), and (3) add some labels (i.e., –OH) in the figure (the revised figure should be information as the reference: https://www.tandfonline.com/doi/abs/10.1080/00986445.2017.1336090)

Response 3: (1) the word “peak” (i.e., “The peaks at 3330 and 2890 cm-1”) was changed to “band” (2) we revised the X-axis of the infrared spectra from 4000 to 510 cm-1 and (2) we added some labels in the figure 6.

Point 4: The authors need to use the non-linear method, the application of the linear method for calculating the parameters of kinetics and isotherm are not currently accepted. The authors should also provide the complete adsorption isotherm (the plot qe vs Ce) that are very important (not as Figure 8). The plot of adsorption kinetics need to presented as qt vs. time (not as Figure 8)

Response 4: I agree with you, the linear method is not accepted in the case of our study. The non-linear method was used for calculating the parameters of kinetics and isotherm. The adsorption isotherm plot of qe vs Ce is presented in Figure 7 and the kinetics plot of qt vs. time is presented in figure 8.

Point 5: The authors only used R2 to find the fit model that are not enough, please use adj-R2 and some statistics such as red-χ2, SD, BIC (https://www.mdpi.com/2073-4441/15/6/1231)

Response 5: The statistical parameters of kinetic and isotherm studies are added in materials and methods section (Line 108-115) and discussion section (Line 304-309) and Tables 6 and 8.

Point 6: For the Thermodynamic parameters, this method applied is not suitable, you cannot defined the standard states in this method (please read the paper: https://www.hindawi.com/journals/ast/2022/5553212/). It is also impossible to obtain a very high ΔG° and ΔH° values (kJ/mol) for this adsorption process. I suggested two ways for this section: (1) adding the adsorption isotherm at least three temperatures and applied the adsorption equilibrium constant (i.e., From KL of the Langmuir model) as the suggested paper (https://www.sciencedirect.com/science/article/pii/S2213343721016511) for recalculating the thermodynamic parameters and increased some discussion, or (2) removing this section from the revised manuscript.

Response 6: Based on your relevant remark and suggested paper, the thermodynamic parameters is recalculated. The correct result with three temperature values is presented in figure 9. Furthermore, the discussion and interpretation is added (Line 366-377).

Point 7: This is unclear. The adsorption mechanisms. The authors should add the FTIR of the material after adsorption. The zeta potentials of the material before and after adsorption (of the pHPZC of the material before and after adsorption). Those data will help to and explain the adsorption mechanism convincingly.

Response 7: I agree with your remark. The data of material FTIR after adsorption and the calculation of potential zeta of material before and after adsorption will dramatically help to enrich the discussion and explain the adsorption mechanism. In fact, at the end of our adsorption experiment, it was not possible to access the FTIR instrument. The potential zeta of sugarcane bagasse is not determined as we do not have the necessary adsorbent to conduct the experiment. Indeed, we are relying on data from the literature concerning bagasse pHPZC to explain the behavior pH after adsorption (Line 217-223).

Point 8: Please recheck the data, if there are any data that were published previously, the authors should give suitable citations.

Response 7: We checked previously the research that is published for the use of sugarcane bagasse as a powerful adsorbent, and we cited some research in the introduction and discussion section of the article that gives information about adsorbent capacity and effluent treated by this adsorbent.
